# Emerging low-cloud feedback and adjustment in global satellite observations

Paulo Ceppi<sup>1</sup>, Sarah Wilson Kemsley<sup>2,3</sup>, Hendrik Andersen<sup>4,5</sup>, Timothy Andrews<sup>6,7</sup>, Ryan J. Kramer<sup>8</sup>, Peer Nowack<sup>4,9</sup>, Casey J. Wall<sup>10,11</sup>, and Mark D. Zelinka<sup>12</sup>

Correspondence: Paulo Ceppi (p.ceppi@imperial.ac.uk)

Abstract. From mid-2003 to mid-2024, a decrease in low-cloud amount enhanced the absorption of solar radiation by  $0.22 \pm 0.07 \,\mathrm{W\,m^{-2}\,decade^{-1}}$  ( $\pm 1\sigma$  range), accelerating the energy imbalance trend during that period ( $0.44 \,\mathrm{W\,m^{-2}\,decade^{-1}}$ ). Through controlling factor analysis, here we show that the low-cloud trend is due to a combination of cloud feedback and adjustments to aerosols and greenhouse gases (respectively  $0.07 \pm 0.01$ ,  $0.06 \pm 0.01$ , and  $0.05 \pm 0.03 \,\mathrm{W\,m^{-2}\,decade^{-1}}$ ), which jointly account for 82% of the trend. The contribution of natural climate variability is weak but uncertain ( $0.03 \pm 0.07 \,\mathrm{W\,m^{-2}\,decade^{-1}}$ ), owing to a poorly constrained trend in boundary-layer inversion strength. Importantly, the observed low-cloud radiative trend lies well within the range of values simulated by contemporary global climate models under conditions close to present day. Any systematic model error in the representation of present-day global energy imbalance trends is thus likely to originate in processes other than low clouds.

#### 10 1 Introduction

Earth's energy imbalance, the difference between absorbed shortwave and outgoing longwave radiation at the top of the atmosphere, is a key indicator of climate change. This energy imbalance is currently increasing under the combined effect of a strengthening positive greenhouse gas forcing and a weakening negative aerosol forcing (Forster et al., 2021; Loeb et al., 2021; Kramer et al., 2021; Hodnebrog et al., 2024; Forster et al., 2025). However, radiative budget trends are also influenced by processes of rapid adjustment and climate feedback, as well as natural climate variability (Raghuraman et al., 2021; Andrews et al., 2022; Raghuraman et al., 2023).

<sup>&</sup>lt;sup>1</sup>Department of Physics, Imperial College London, London, United Kingdom

<sup>&</sup>lt;sup>2</sup>Climatic Research Unit, School of Environmental Sciences, University of East Anglia, Norwich, United Kingdom

<sup>&</sup>lt;sup>3</sup>School of Geography and the Environment, University of Oxford, Oxford, United Kingdom

<sup>&</sup>lt;sup>4</sup>Institute of Meteorology and Climate Research Atmospheric Trace Gases and Remote Sensing, Karlsruhe Institute of Technology, Karlsruhe, Germany

<sup>&</sup>lt;sup>5</sup>Institute of Photogrammetry and Remote Sensing, Karlsruhe Institute of Technology, Karlsruhe, Germany

<sup>&</sup>lt;sup>6</sup>Met Office Hadley Centre, Exeter, United Kingdom

<sup>&</sup>lt;sup>7</sup>School of Earth and Environment, University of Leeds, Leeds, United Kingdom

<sup>&</sup>lt;sup>8</sup>Geophysical Fluid Dynamics Laboratory, NOAA, Princeton, NJ, USA

<sup>&</sup>lt;sup>9</sup>Institute of Theoretical Informatics, Karlsruhe Institute of Technology, Karlsruhe, Germany

<sup>&</sup>lt;sup>10</sup>Department of Meteorology, Stockholm University, Stockholm, Sweden

<sup>&</sup>lt;sup>11</sup>Bolin Centre for Climate Research, Stockholm University, Stockholm, Sweden

<sup>&</sup>lt;sup>12</sup>Lawrence Livermore National Laboratory, Livermore, CA, United States

Global satellite observations reveal a rapidly increasing global energy imbalance since the early 2000s, seemingly faster than simulated by contemporary coupled global climate models (Olonscheck and Rugenstein, 2024; Mauritsen et al., 2025), but the causes of this discrepancy are unclear. Previous work has highlighted the important contribution of decreased shortwave (SW) reflection by clouds (Stephens et al., 2022; Raghuraman et al., 2023; Tselioudis et al., 2025), and particularly low clouds (Loeb et al., 2024b; Goessling et al., 2025), to the energy imbalance trends. Recent decades have seen a conjunction of factors expected to reduce low-cloud SW reflection: weakening aerosol forcing (Kramer et al., 2021; Yuan et al., 2022, 2024; Gettelman et al., 2024; Zhang et al., 2025); positive rapid adjustments to increasing greenhouse gas forcing (Gregory and Webb, 2008; Andrews and Forster, 2008; Andrews et al., 2012); and positive low-cloud feedback (Ceppi et al., 2017; Zelinka et al., 2020; Myers et al., 2021; Ceppi et al., 2024). Meanwhile, natural climate variability can cause large decadal SW low-cloud trends of either sign, particularly via the sea-surface temperature "pattern effect" (Zhou et al., 2016; Andrews et al., 2022). The relative importance of these various drivers for the recent low-cloud radiative trends remains to be quantified.

Here we apply cloud-controlling factor analysis (e.g., Scott et al., 2020; Myers et al., 2021; Ceppi and Nowack, 2021; Ceppi et al., 2024) to interpret the low-cloud radiative trends from July 2003 to June 2024, using state-of-the-art global satellite observations from the Clouds and the Earth's Radiant Energy System (CERES) project. We find that the low-cloud trend is driven primarily by a combination of low-cloud feedback and adjustments to aerosol and greenhouse gas (GHG) forcing. Furthermore, the observed low-cloud trend lies well within the range of global climate model simulations, suggesting that any model error in energy imbalance trends likely originates in processes other than low clouds.

#### 2 Observed radiative trends

From July 2003 to June 2024, Earth's global energy imbalance N as observed by CERES Energy Balanced and Filled (CERES-EBAF) increased at an average rate of  $0.44 \,\mathrm{W\,m^{-2}\,decade^{-1}}$  (Fig. 1a) – a value consistent with other findings based on similar analysis periods (e.g., Hodnebrog et al., 2024; Loeb et al., 2024b; Mauritsen et al., 2025). SW low-cloud anomalies ( $R_{\mathrm{SWlow}}$ , Appendices A1–A2), calculated from the CERES Flux-By-Cloud-Type (CERES-FBCT) product, made a large contribution amounting to half of this decadal trend, in addition to explaining a substantial portion of inter-annual variations in N (Fig. 1a; correlation coefficient r=0.82).

The global  $R_{\rm SWlow}$  trend is driven by changes across the Northern Hemisphere ocean basins, Europe, the Southeast Indian Ocean, and the South Atlantic, with opposing contributions mainly in the tropical Southeast Pacific (Fig. 2a). Given that low clouds exhibit a positive radiative sensitivity to surface temperature ( $T_{\rm sfc}$ ), and a negative sensitivity to estimated inversion strength (EIS, defined in Appendix A1; Fig. A1), the spatial pattern of the  $R_{\rm SWlow}$  trend appears qualitatively consistent with the observed changes in  $T_{\rm sfc}$  and EIS (Fig. 2d–e): regions of positive  $R_{\rm SWlow}$  trends coincide with regions of increasing  $T_{\rm sfc}$ , and the area of negative  $R_{\rm SWlow}$  change in the tropical Southeast Pacific corresponds to an area of positive EIS change and near-zero  $T_{\rm sfc}$  trend. The low-cloud radiative trends described here agree in magnitude and meridional structure with those reported in Fig. 9b of Loeb et al. (2024b) for the period July 2002 to December 2022.

## Global radiative trends

Figure 1. (a–c) Timeseries of global radiative anomalies: (a) CERES-EBAF net radiative imbalance N (green) and CERES-FBCT  $R_{\rm SWlow}$  (black); (b) CERES-FBCT  $R_{\rm SWlow}$  (black) and reconstructed timeseries (grey); (c) reconstructed CERES-FBCT  $R_{\rm SWlow}$  (grey) and  $R_{\rm SWlow}$  from CMIP6 models (Table A1), emulating extended *amip* simulations (dark orange). The actual  $R_{\rm SWlow}$  from *amip* simulations up to December 2014 is shown in light orange. (d) CERES-FBCT actual and reconstructed trends, *amip* emulated trend, and contributions to the CERES-FBCT reconstructed trend from cloud feedback, aerosol adjustment, greenhouse gas (GHG) adjustment, and unforced climate variability. Thick bars denote central estimates, while thin bars provide  $\pm 1\sigma$  ranges. Timeseries show monthly anomalies from the time mean, smoothed with a 12-month centred running mean; coloured dashed lines represent linear fits to the corresponding timeseries, with decadal trend values shown in the legends. Values are near-global averages (60 °S to 60 °N), scaled to the global area, except for N which uses global data.

The  $R_{\rm SWlow}$  changes also agree well with the changes in low-cloud amount, both locally and globally (Fig. 2a–c); low-cloud amount decreases globally by  $0.05\%\,\rm decade^{-1}$  during the analysis period. Consistent with this, nearly all of the  $R_{\rm SWlow}$  trend is associated with decreasing cloud amount, as opposed to decreasing optical depth (respectively 0.21 and  $0.05\,\rm W\,m^{-2}\,decade^{-1}$ ; not shown).

Our study period exhibits a substantial near-global  $T_{\rm sfc}$  increase of  $0.20\,{\rm K\,decade}^{-1}$ , corresponding to an increase of  $0.44\,{\rm K}$  over 21 years. This suggests that a substantial fraction of the  $R_{\rm SWlow}$  increase may reflect an emerging low-cloud feedback in satellite observations. However, aerosol adjustment is likely to have also played a role, especially in the Northern Hemisphere,

# Decadal trends, July 2003 to June 2024

Figure 2. Observed decadal trends in (a) low-cloud radiative effect  $R_{\rm SWlow}$ , (b) low-cloud amount  $C_{\rm low}$  (note the inverted colourbar), (d) surface temperature  $T_{\rm sfc}$ , and (e) estimated inversion strength EIS (lower-tropospheric stability LTS over land). (c,f) Near-global timeseries of the same quantities, averaged from  $60\,^{\circ}{\rm S}$  to  $60\,^{\circ}{\rm N}$  and scaled to the global area. Coloured dashed lines represent linear fits to the corresponding timeseries, with decadal trend values shown in the legends. The timeseries show monthly anomalies from the time-mean, smoothed with a 12-month centred running mean. Trend maps for other controlling factors are shown in Fig. A2.

as are GHG adjustments and unforced climate variability. To distinguish between these drivers, we employ cloud-controlling factor (CCF) analysis (Appendix A3) along with global climate model estimates of the  $R_{\rm SWlow}$  GHG adjustment.

## 3 Drivers of the low-cloud radiative trend

65

The reconstruction method (Appendix A3) overall performs well in reproducing not only the inter-annual variability in global  $R_{\rm SWlow}$ , but also its decadal trend (Fig. 1b,d), even though the latter was removed prior to CCF analysis. Some extrema are however underestimated, particularly around years 2009, 2014 and 2019. Discrepancies may arise for several reasons: inaccuracies in the CCF method, observational error in the CCFs or the cloud-radiative anomalies, stochastic cloud variability that is unaccounted for by the CCFs, or errors in the climate model-based estimate of the GHG adjustment. Despite these limitations, the global  $R_{\rm SWlow}$  trend is reproduced nearly perfectly, being underestimated by just  $0.01 \, {\rm W \, m^{-2} \, decade^{-1}}$ .

The observed  $R_{\rm SWlow}$  trend is also reasonably well reproduced spatially (Fig. 3a–b), with the CCF reconstruction correctly capturing the distribution of positive and negative anomalies, and a pattern correlation coefficient of 0.70. Discrepancies are found mainly in Northern Hemisphere ocean basins, as well as in Southern Hemisphere stratocumulus regions (Fig. 3c). We can

Figure 3. Maps of CERES-FBCT  $R_{\rm SWlow}$  trends, decomposed into contributions. (a) Actual  $R_{\rm SWlow}$  trend, (b) reconstructed trend (sum of panels d–g), (c) difference (a minus b), and contributions from (d) cloud feedback, (e) aerosol adjustment, (f) greenhouse gas (GHG) adjustment, and (g) unforced variability.

further validate the method by treating each available CMIP6 *historical* model simulation (Appendix A1) as an observation, and showing that the reconstructed trends agree closely with the actual values (Fig. 4).

We next assess the contributions of cloud feedback, adjustment to aerosols and GHG, and unforced climate variability to the observed  $R_{\rm SWlow}$  trend (Appendix A4–A5). We identify cloud feedback as the leading contribution, explaining a third of the observed global  $R_{\rm SWlow}$  trend (0.07  $\pm$  0.01 W m<sup>-2</sup> decade<sup>-1</sup>,  $\pm$ 1 $\sigma$  range; Fig. 1d). This contribution is largest in tropical subsidence and midlatitude regions (Fig. 3d), consistent with previous observational assessments (Myers et al., 2021; Ceppi

80

Figure 4. Comparison of actual and reconstructed trends in  $R_{\rm SWlow}$ . Blue symbols denote trends from individual CMIP6 models in the historical simulations during January 1995 to December 2014. The black circle shows CERES-FBCT values for July 2003 to June 2024, with error bars denoting  $\pm 1\sigma$  confidence intervals. The one-to-one line is shown in solid black.

et al., 2024). Normalised by the observed warming rate, this implies a low-cloud feedback of  $0.37 \, \mathrm{W \, m^{-2} \, K^{-1}}$ , within the range assessed by Ceppi et al. (2024).

The next largest effects are adjustments to aerosol and GHG. Aerosols contribute  $0.06\pm0.01\,\mathrm{W\,m^{-2}\,decade^{-1}}$ , primarily from changes in the Northern Hemisphere (Fig. 3e). GHG adjustment has an impact similar to aerosols  $(0.05\pm0.03\,\mathrm{W\,m^{-2}\,decade^{-1}})$ , but with a much more uniform spatial pattern (Fig. 3f). Taken together, low-cloud feedback and adjustment (i.e. the forced low-cloud response) account for around 82% of the trend  $(0.18\pm0.03\,\mathrm{W\,m^{-2}\,decade^{-1}}$ ; Fig. 1d).

Accordingly, the contribution of unforced climate variability is relatively small at  $0.03\,\mathrm{W\,m^{-2}\,decade^{-1}}$ . This is likely coincidental and specific to the phase of climate variability for the time period considered, given the known large decadal variations in low-cloud feedback (Zhou et al., 2016; Kawaguchi and Ceppi, 2025). While small in the global mean, the unforced variability component exhibits a large uncertainty ( $\pm 0.07\,\mathrm{W\,m^{-2}\,decade^{-1}}$ ) and is regionally dominant (Fig. 3g). In a global-mean sense, the magnitude and sign of the unforced variability contribution depend entirely on the choice of EIS dataset (Appendices A5–A6): across the 20 ensemble members, the unforced trend correlates nearly perfectly with the trend of the EIS contribution to  $R_{\mathrm{SWlow}}$  (r = 0.99; not shown). This EIS trend uncertainty also dominates the spread in the total reconstructed  $R_{\mathrm{SWlow}}$  trend (r = 0.91; Fig. A4). Importantly, however, the choice of EIS or aerosol CCF dataset has comparatively little impact on the forced trend contribution and its components (Fig. 1d).

## 4 Can global climate models simulate the low-cloud trend?

Given prior findings that CMIP6 models may underestimate the recent energy imbalance trend (Olonscheck and Rugenstein, 2024; Mauritsen et al., 2025), it is reasonable to ask whether low clouds may account for some or all of the discrepancy. Per Eq. A1, such a discrepancy could arise for two, not mutually exclusive reasons: 1) models are unable to simulate the CCF changes observed during the period of study (the dX term in Eq. A1); 2) models misrepresent the cloud response to the observed CCF changes (the  $\Theta$  term). It is additionally possible that models underestimate the low-cloud adjustment to GHG forcing, but we are unable to assess that possibility here.

To evaluate model performance, we compare the observed  $R_{\rm SWlow}$  trend with that calculated from *amip* and *historical* simulations. Comparing with *amip* minimises differences in CCF trends between models and observations, thus highlighting the role of the cloud-radiative sensitivities ( $\Theta$ ). By contrast, the comparison with *historical* will additionally include more substantial differences in CCF trends.

The *amip* and *historical* experiments in CMIP6 end in December 2014. Hence for the comparison with *amip* simulations, we restrict ourselves to the overlapping period July 2003 to December 2014 (Fig. 1c, light orange). Additionally, however, we exploit CCF analysis to generate synthetic *amip*  $R_{\rm SWlow}$  timeseries extended up to June 2024, by convolving each model's own CCF sensitivities (calculated from independent *historical* simulations) with the observed CCF anomalies (Fig. 1c, dark orange). In either case, the multi-model *amip* results agree well with CERES observations, in terms of both year-to-year fluctuations and the long-term trend (Fig. 1c). The reconstructed *amip* trend is slightly weaker than that reconstructed from observed sensitivities (0.17 versus  $0.21\,\mathrm{W\,m^{-2}\,decade^{-1}}$ ), although this is highly model-dependent ( $\pm 1\sigma$  inter-model range -0.01 to  $+0.36\,\mathrm{W\,m^{-2}\,decade^{-1}}$ ). Overall, the comparison suggests that the observed  $R_{\rm SWlow}$  trend lies well within the range of what contemporary climate models would simulate if extended *amip* simulations were available.

Now turning to the *historical* experiment, since the CCF changes are not constrained by observed sea-surface temperatures, a perfect time overlap is not necessary and hence we use the most recent 20-year period January 1995 to December 2014 for our comparison. Here again, the CERES  $R_{\rm SWlow}$  trend lies fully within the range of model-simulated trends, if towards the upper end of the distribution (Fig. 4). The two models simulating stronger positive trends – two versions of the UK Met Office model, HadGEM3-GC31-LL and UKESM1-0-LL – are also at the upper end of the CMIP6 range in terms of their low-cloud feedback (Ceppi et al., 2024, their Table S2).

A limitation of our analysis is that the results in Figs. 1c and 4 are based on a limited set of CMIP6 models, including several high-sensitivity models (Table A1), meaning we cannot reliably assess any systematic model bias. Furthermore, the trends in Fig. 4 are likely strongly influenced by the phase of natural climate variability in individual realizations, and hence a quantitative comparison between models and observations would require the use of large ensembles, as in Olonscheck and Rugenstein (2024). Besides, forced CCF trends during 1995–2014 likely differ slightly from those acting in the observational period 2003–2024. These limitations notwithstanding, the results indicate that contemporary global climate models are able to simulate  $R_{\rm SWlow}$  trends similar to those observed, whether or not they are constrained to follow the specific phase of observed climate variability.

#### 5 Conclusions

Earth's energy imbalance grew by  $0.44\,\mathrm{W\,m^{-2}\,decade^{-1}}$  between July 2003 and June 2024. Over this 21-year period, this amounts to an increase of  $0.92\,\mathrm{W\,m^{-2}}$ , as large as the mean imbalance itself (Mauritsen et al., 2025) and potentially in excess of the rate simulated by contemporary global climate models (Olonscheck and Rugenstein, 2024). Using cloud-controlling factor (CCF) analysis, combined with climate model-derived estimates of rapid adjustments to greenhouse gas (GHG) forcing, we show that shortwave (SW) radiative changes by low clouds substantially contribute to this energy imbalance increase, at  $0.22\,\mathrm{W\,m^{-2}\,decade^{-1}}$ . The low-cloud trends, in turn, are driven to a comparable extent by low-cloud feedback, sulfate aerosol adjustment, and GHG adjustment, which jointly account for around 82% of the trend. This leaves only a minor role for natural climate variability, although our estimate is subject to a substantial uncertainty related to trends in 700 hPa temperature and thus EIS (Appendix A6). Our assessment of the natural variability component additionally depends on the assumption that climate models realistically represent forced CCF trends.

A comparison with CMIP6 *amip* and *historical* global climate model simulations reveals that, for either experiment, observed SW low-cloud trends lie well within the range of simulated trends. In particular, emulated *amip* trends agree well with CERES observations in a multi-model mean sense. Based on this comparison, the observed substantial low-cloud radiative trend cannot be interpreted as evidence of an unexpectedly strong low-cloud feedback that climate models are systematically missing (Goessling et al., 2025). A caveat is that the comparison is based on a limited set of climate models (Table A1).

In light of our findings, it remains unclear why state-of-the-art climate models appear to generally underestimate recent trends in global energy imbalance (Schmidt et al., 2023; Olonscheck and Rugenstein, 2024; Hodnebrog et al., 2024; Mauritsen et al., 2025). We propose that further research should quantify and constrain the contributions of processes other than low clouds to the observed and modelled energy imbalance trends, including their decomposition into forcing and radiative response (Raghuraman et al., 2021).

Data availability. The HadGEM3 EIS amip data used in Fig. A3b will be made available online prior to publication. All other datasets used here are freely available online: CERES-EBAF and CERES-FBCT (NASA/LARC/SD/ASDC, 2020, 2022, 2023); ERA5 (Hersbach et al., 2023a, b); MERRA2 (Global Modeling and Assimilation Office (GMAO), 2015a, b); JRA3Q (Japan Meteorological Agency (JMA), 2022); AIRS (AIRS project, 2019); CLIMCAPS (Barnet, 2019); CMIP6 (https://esgf-node.llnl.gov).

## Appendix A: Data and methods

# A1 Data

We use gridded global satellite observations of cloud amount and top-of-atmosphere radiative fluxes from the CERES Flux-By-Cloud-Type (CERES-FBCT) product (Sun et al., 2022), combining Edition 4A fluxes from Terra and Aqua available up to February 2023, and Edition 1B fluxes from NOAA-20 for the remainder of the analysis period, taking care to minimise

discontinuities between the two products (Appendix A2). Since CERES-FBCT fluxes are provided as a function of cloud-top pressure, we can isolate the contribution of low clouds (cloud-top pressure greater than 680 hPa) to radiation budget changes (Appendix A2). Note that CERES-FBCT provides cloud-radiative effect rather than true cloud-induced radiative anomalies, and hence the fluxes are subject to cloud masking effects (Soden et al., 2004); these are however expected to be much smaller for SW than longwave (LW) fluxes, particularly since we exclude regions poleward of  $60^{\circ}$  where surface albedo changes are largest (Raghuraman et al., 2023). This, combined with the fact the LW effects of low clouds are small, motivates our focus on SW anomalies, hereafter denoted  $R_{\rm SWlow}$  (defined positive downward).

To provide context for the low-cloud trends, we additionally use estimates of the net top-of-atmosphere radiative budget (hereafter *N*) from CERES Energy Balanced and Filled (CERES-EBAF) observations, Edition 4.2 (Loeb et al., 2018). When summed over all cloud types, the cloud-radiative changes diagnosed from CERES-FBCT provide a close match to those obtained from CERES-EBAF (Loeb et al., 2024b), making CERES-FBCT ideally suited for quantifying the contributions of different cloud types to changes in Earth's energy imbalance.

We consider seven meteorological drivers of cloud property changes, hereafter cloud-controlling factors (CCFs): surface temperature,  $T_{\rm sfc}$ ; estimated inversion strength, EIS (Wood and Bretherton, 2006); 700 hPa relative humidity, RH<sub>700</sub>; 700 hPa pressure velocity,  $\omega_{700}$ ; sea-surface temperature (SST) advection, SSTadv; near-surface wind speed, WS; and the base-10 logarithm of sulfate aerosol optical depth at 550 nm, log(AOD). Note that over land, instead of EIS we use the simpler metric of lower-tropospheric stability (Klein and Hartmann, 1993), as the EIS metric involves assumptions that would only hold over the ocean surface.

Controlling factor data are taken from the the European Centre for Medium-Range Weather Forecasts (ECMWF) reanalysis, version 5 (ERA5; Hersbach et al., 2020) and the Modern-Era Retrospective analysis for Research and Applications, version 2 (MERRA2; Gelaro et al., 2017). Because EIS trends are sufficiently uncertain as to impact the results (Appendix A6, Figs. A3–A4; see also Myers et al., 2023), we include an additional three independent estimates: the JRA3Q reanalysis (Kosaka et al., 2024), the Atmospheric Infrared Sounder (AIRS) satellite product, version 7 (Aumann et al., 2003), and the Community Long-term Infrared Microwave Combined Atmospheric Product System (CLIMCAPS) satellite product, version 2 (Smith and Barnet, 2023). Note that CLIMCAPS uses measurements from AIRS, but combined with additional instruments and processed with a different algorithm. Since EIS trend discrepancies primarily depend on the evolution of 700 hPa temperature ( $T_{700}$ ; not shown), we combine AIRS and CLIMCAPS  $T_{700}$  with ERA5 surface temperature and sea-level pressure to calculate EIS, while in other cases the values are taken from the corresponding reanalysis product. For log(AOD), we consider two reanalysis products: the Copernicus Atmosphere Monitoring Service (CAMS), and MERRA2. The dependence of CCF trends on the choice of dataset is illustrated in Figs. A3–A4 and discussed in Appendix A6.

We perform a similar analysis with data from global climate model simulations available from the Coupled Model Intercomparison Project phase 6 (CMIP6; Eyring et al., 2016).  $R_{\rm SWlow}$  is calculated using International Satellite Cloud Climatology Project (ISCCP; Rossow and Schiffer, 1999) satellite simulator output (Bodas-Salcedo et al., 2011) convolved with cloud-radiative kernels (Zelinka et al., 2012), accounting for effects of obscuration by non-low clouds (Zelinka et al., 2025). We

define low-cloud radiative anomalies using the lowest three ISCCP simulator levels (instead of two for CERES-FBCT) owing to a known ISCCP bias (Myers et al., 2021; Ceppi et al., 2024).

CMIP6 CCF sensitivities are calculated from the final 20 years of the CMIP6 *historical* experiment, January 1995 to December 2014. We additionally use data from the *amip*, *ssp245*, *piClim-control*, and *piClim-ghg* experiments; the models, variables and time periods used for each experiment are summarised in Table A1.

All observed and simulated fields are in monthly resolution, and are remapped to a common  $5^{\circ} \times 5^{\circ}$  grid in longitude and latitude prior to statistical analysis.

## A2 Calculation of CERES-FBCT low-cloud radiative effect

195 CERES-FBCT data consists of clear-sky radiative flux  $R_{\rm clr}$ , cloudy-sky radiative flux  $R_{\rm cld}$ , and cloud amount. The latter two are partitioned according to seven cloud-top pressure (p) bins and six cloud optical depth  $(\tau)$  bins. Following previous literature, we categorise clouds in the lowest two bins (p > 680 hPa) as low clouds. We use only the shortwave (SW) component of the fluxes, denoted  $R_{\rm SWclr}$  and  $R_{\rm SWcld}$ .

For the passive satellite retrievals used here, month-to-month variations in low-cloud amount L could arise simply because of changes in the amount of upper-level clouds U obscuring lower-level clouds. To isolate the contribution of low clouds, we make an assumption of random overlap between low and upper-level clouds. For each  $(p,\tau)$  bin, we thus define non-obscured low-cloud amount  $L_n$  as

$$L_n(p,\tau) = \frac{L(p,\tau)}{1-U}.$$

To obtain low-cloud radiative fluxes, we define the difference  $K_{\rm SW}(p,\tau)=R_{\rm SWcld}(p,\tau)-R_{\rm SWclr}$ , which quantifies how top-of-atmosphere radiation changes in the presence versus absence of clouds for each month and p- $\tau$  bin, similar to a cloud-radiative kernel (Zelinka et al., 2012). We can then calculate the low-cloud contribution to top-of-atmosphere radiative flux,  $R_{\rm SWlow}$ , by convolving  $K_{\rm SW}$  with non-obscured low-cloud amount  $L_n$ , and summing over the lowest two p bins and all six  $\tau$  bins:

$$R_{\text{SWlow}} = (1 - \bar{U}) \sum_{p=1}^{2} \sum_{\tau=1}^{6} K_{\text{SW}} L_n.$$

Note that this calculation uses absolute low-cloud amount values (rather than anomalies) and thus provides absolute radiative flux contributions (the contribution of low clouds to SW cloud-radiative effect).

As a final step before calculating the CCF sensitivities, we normalise  $R_{\rm SWlow}$  to annual-mean insolation conditions, as seasonal changes in insolation affect  $R_{\rm SWlow}$  in the absence of any physical cloud changes, thus constituting a confounding factor. We thus define  $\tilde{R}_{\rm SWlow} = R_{\rm SWlow} \times \bar{S}/S$ , where S is top-of-atmosphere downward SW and the overbar denotes the annual-mean climatology. We then use  $\tilde{R}_{\rm SWlow}$  in the ridge regression to calculate the CCF sensitivities.

CERES-FBCT fluxes come from two products covering different time periods. The Edition 4A product is based on retrievals from the Terra and Aqua satellites during September 2002 to February 2023; the Edition 1B product instead uses retrievals

**Table A1.** List of CMIP6 models used in the analysis, with the corresponding experiments and time periods used. A cross  $(\times)$  denotes available data.

|                 | Experiments |                |                            |                       |
|-----------------|-------------|----------------|----------------------------|-----------------------|
|                 | historical  | amip           | piClim-control, piClim-ghg | historical,<br>ssp245 |
| Model name      | 1995–2014   | 7/2003–12/2014 | Years 2–30                 | 7/2003–6/2024         |
| ACCESS-CM2      |             |                |                            | ×                     |
| ACCESS-ESM1-5   |             |                |                            | ×                     |
| AWI-CM-1-1-MR   |             |                |                            | ×                     |
| BCC-CSM2-MR     |             |                |                            | ×                     |
| CAMS-CSM1-0     |             |                |                            | ×                     |
| CanESM5         | ×           |                | ×                          | ×                     |
| CAS-ESM2-0      |             |                |                            | ×                     |
| CMCC-CM2-SR5    |             |                |                            | ×                     |
| CMCC-ESM2       |             |                |                            | ×                     |
| CNRM-CM6-1      | ×           | ×              | ×                          | ×                     |
| CNRM-CM6-1-HR   |             |                |                            | ×                     |
| CNRM-ESM2-1     | ×           | ×              | ×                          | ×                     |
| EC-Earth3-Veg   |             |                |                            | ×                     |
| FGOALS-f3-L     |             |                |                            | ×                     |
| GFDL-CM4        | ×           |                | ×                          | ×                     |
| GFDL-ESM4       |             |                |                            | ×                     |
| HadGEM3-GC31-LL | ×           |                | ×                          | ×                     |
| IITM-ESM        |             |                |                            | ×                     |
| INM-CM4-8       |             |                |                            | ×                     |
| INM-CM5-0       |             |                |                            | ×                     |
| IPSL-CM6A-LR    | ×           | ×              | ×                          | ×                     |
| KACE-1-0-G      |             |                |                            | ×                     |
| KIOST-ESM       |             |                |                            | ×                     |
| MIROC-ES2L      |             |                | ×                          | ×                     |
| MIROC6          |             |                | ×                          | ×                     |
| MPI-ESM1-2-HR   |             |                | **                         | ×                     |
| MPI-ESM1-2-LR   |             |                |                            | ×                     |
| MRI-ESM2-0      | ×           | ×              | ×                          | ×                     |
| NESM3           | . •         |                |                            | ×                     |
| UKESM1-0-LL     | ×           |                | ×                          | ×                     |

from NOAA-20 and covers the period May 2018 to July 2024. Loeb et al. (2024a) have documented a discontinuity in CERES-EBAF radiative fluxes between the two satellites, and described a method exploiting the time overlap between the two products to adjust the NOAA-20 fluxes and anchor them to the Terra-Aqua values. Here, we follow the same procedure to merge the two CERES-FBCT products. The procedure is applied to  $R_{\rm SWlow}$ , meaning that we first separately calculate  $R_{\rm SWlow}$  for each of the two products, then apply the method below to combine the  $R_{\rm SWlow}$  values.

Specifically, during the overlap period May 2018 to February 2023, we calculate monthly  $R_{\rm SWlow}$  climatologies at each gridpoint for both products. We then compute the climatology difference for each gridpoint and calendar month, and subtract this difference from the NOAA-20 values over the entire record. This yields a modified CERES-FBCT NOAA-20 product whose  $R_{\rm SWlow}$  climatology during the overlap period is identical to that of the Terra–Aqua product. Finally, we concatenate the Terra–Aqua data up to February 2023 with the modified NOAA-20 data from March 2023 to July 2024, to produce a single record from September 2002 to July 2024.

## A3 Cloud-controlling factor analysis framework

The CCF analysis approach follows Ceppi et al. (2024), with the addition of an aerosol CCF following Wall et al. (2022). Briefly, the SW low-cloud radiative anomalies at each location r,  $dR_{SWlow}(r)$ , are modelled as

$$dR_{\text{SWlow}}(r) \approx \sum_{i=1}^{7} \frac{\partial R_{\text{SWlow}}(r)}{\partial \boldsymbol{X}_{i}(r)} \cdot d\boldsymbol{X}_{i}(r) = \sum_{i=1}^{7} \boldsymbol{\Theta}_{i}(r) \cdot d\boldsymbol{X}_{i}(r), \tag{A1}$$

where  $X_i$  represents one of seven CCFs, and  $\Theta_i$  denotes the sensitivity of  $R_{\rm SWlow}$  to each CCF  $X_i$ . To model  $R_{\rm SWlow}$  at each point r, we use CCF information from a  $5\times 5$  regional domain (in gridbox units) centred around r; hence  $X_i$  and  $\Theta_i$  are spatial vectors, denoted by bold typeface in Eq. A1. The sensitivities  $\Theta_i$  are calculated via ridge regression, where all variables have been deseasonalised, and the CCF predictors have been standardised. Different from Ceppi et al. (2024), we additionally linearly detrend all variables prior to calculating the regressions; this ensures that the trend we are attempting to explain is not part of the training data. We train on the 20-year period January 2003 to December 2022; the large positive  $R_{\rm SWlow}$  anomaly in 2023 is thus predicted entirely out-of-sample (Fig. 1b). The resulting cloud-radiative sensitivities (Fig. A1 are in agreement with previous findings (Scott et al., 2020; Myers et al., 2021; Ceppi et al., 2024).

#### A4 Trend decomposition method

We decompose the reconstructed, observed  $R_{\rm SWlow}$  trend,  $\frac{\mathrm{d}R_{\rm SWlow}}{\mathrm{d}t}\big|_{\rm rec}$ , into a forced component ( $\frac{\mathrm{d}R_{\rm SWlow}}{\mathrm{d}t}\big|_{\rm for}$ ) and a contribution due to unforced climate variability ( $\frac{\mathrm{d}R_{\rm SWlow}}{\mathrm{d}t}\big|_{\rm unf}$ ). We assume that the forced component is itself driven by a combination of cloud feedback ( $\frac{\mathrm{d}R_{\rm SWlow}}{\mathrm{d}t}\big|_{\rm fdbk}$ ) and adjustments to sulfate aerosol ( $\frac{\mathrm{d}R_{\rm SWlow}}{\mathrm{d}t}\big|_{\rm aer}$ ) and GHG ( $\frac{\mathrm{d}R_{\rm SWlow}}{\mathrm{d}t}\big|_{\rm GHG}$ ):

$$\frac{dR_{\text{SWlow}}}{dt}\Big|_{\text{rec}} = \frac{dR_{\text{SWlow}}}{dt}\Big|_{\text{for}} + \frac{dR_{\text{SWlow}}}{dt}\Big|_{\text{unf}}$$

$$= \frac{dR_{\text{SWlow}}}{dt}\Big|_{\text{fdbk}} + \frac{dR_{\text{SWlow}}}{dt}\Big|_{\text{aer}} + \frac{dR_{\text{SWlow}}}{dt}\Big|_{\text{GHG}} + \frac{dR_{\text{SWlow}}}{dt}\Big|_{\text{unf}}.$$
(A2)

**Figure A1.** Maps of the cloud-radiative sensitivities to each controlling factor, representing the average sensitivity across all ensemble members (Appendix A5). While the sensitivity maps are four-dimensional (one regional map per target gridbox), for visualization purposes we sum over regional domains to yield two-dimensional maps.

The above contributions are calculated from CCF analysis, except for GHG adjustment (described below). We first describe how the CCF trends are decomposed, before explaining the calculation of the radiative trend components.

## A4.1 CCF trend decomposition

We first decompose the observed trends of the six meteorological (i.e. non-aerosol) CCFs into forced and unforced components, where the forced component is in turn driven by a combination of SST changes and adjustments to GHG forcing:

$$\frac{dX}{dt} = \frac{dX}{dt} \Big|_{for} + \frac{dX}{dt} \Big|_{unf}$$

$$= \frac{dX}{dt} \Big|_{SST} + \frac{dX}{dt} \Big|_{GHG} + \frac{dX}{dt} \Big|_{unf}, \tag{A3}$$

where X refers to a meteorological CCF. Different from the *meteorological* CCF trends, we treat the observed *aerosol* CCF trend as entirely forced:  $\frac{d \log(AOD)}{dt} = \frac{d \log(AOD)}{dt} \Big|_{for}$ .

We use climate model simulations to calculate the terms on the right-hand side of Eq. A3, as follows:

265

280

First, we estimate the forced meteorological CCF trend component, dX/dt | for, as the CMIP6-mean trend in the combined historical and ssp245 experiments from July 2003 to June 2024, based on output from 30 models (Table A1). We then scale the model-mean trends by 1.07, corresponding to the ratio of observed to modelled global warming rate; this assumes that the rate of global warming is forced on this 21-year timescale, and that the CCF forced responses scale with global warming.

- Next, for the GHG adjustment-induced trend, dX/dt | GHG, we use the CMIP6-mean CCF response from experiments piClim-ghg and piClim-control (Table A1). This represents the CCF adjustment as of year 2014, relative to the preindustrial control. To assess the corresponding trend contribution, we scale the year 2014 CCF anomalies according to the trend of the ratio of greenhouse gas forcing relative to year 2014, using radiative forcing values from Forster et al. (2025).
  - The SST-mediated CCF trend is then obtained as  $\frac{\mathrm{d}X}{\mathrm{d}t}\big|_{\mathrm{SST}} = \frac{\mathrm{d}X}{\mathrm{d}t}\big|_{\mathrm{for}} \frac{\mathrm{d}X}{\mathrm{d}t}\big|_{\mathrm{GHG}}$ .
  - Finally, the unforced CCF trend is calculated as  $\frac{dX}{dt}\Big|_{unf} = \frac{dX}{dt} \frac{dX}{dt}\Big|_{for}$ , i.e. the residual observed trend.

The resulting forced and unforced CCF trends are shown in Fig. A2.

## A4.2 Radiative trend decomposition

We associate the aerosol adjustment contribution with the aerosol CCF trend, the cloud feedback contribution with the SSTdriven trend of the six meteorological CCFs, and the unforced variability contribution with the unforced CCF trends:

$$\frac{dR_{\text{SWlow}}}{dt} \Big|_{\text{aer}} \approx \Theta_{\text{log(AOD)}} \cdot \frac{d\log(\text{AOD})}{dt},$$

$$\frac{dR_{\text{SWlow}}}{dt} \Big|_{\text{fdbk}} \approx \sum_{i=1}^{6} \Theta_{i} \cdot \frac{dX_{i}}{dt} \Big|_{\text{SST}},$$

$$\frac{dR_{\text{SWlow}}}{dt} \Big|_{\text{unf}} \approx \sum_{i=1}^{6} \Theta_{i} \cdot \frac{dX_{i}}{dt} \Big|_{\text{unf}},$$
(A4)

where  $\Theta$  is the cloud-radiative sensitivity and we have ignored spatial indices for readability.

For GHG adjustments, instead of using CCF analysis as per Eq. A4, we rely entirely on model simulations: contrary to theoretical and modelling evidence of a moderate positive  $R_{\text{SWlow}}$  adjustment (Andrews et al., 2012), CCF analysis predicts a weakly negative response (not shown). We proceed in the exact same way as for the calculation of  $\frac{dX}{dt}|_{\text{GHG}}$  above, but using  $R_{\text{SWlow}}$  instead of CCF X.

Several assumptions and limitations apply to our trend decomposition method. First, we implicitly assume that the observed global-mean temperature trend is a forced signal, and that the CCF changes congruent with global-mean temperature represent a response to the forcing, i.e. a feedback. The method further assumes that CMIP6 models realistically capture the forced response pattern of SST and other variables, as well as the cloud adjustment to GHG. Evidence suggests models may be biased in their representation of present-day forced response patterns (e.g., Wills et al., 2022; Simpson et al., 2025), so the

Figure A2. Maps of the decadal trends in each cloud-controlling factor (CCF), representing average values across all CCF datasets (Fig. A3).

290

numbers based on our decomposition should be interpreted with caution. Note that, if the real-world forced CCF trends were closer to observed than to CMIP6-simulated trends, our decomposition method would yield a smaller unforced  $R_{\rm SWlow}$  trend component.

# 285 A5 Uncertainty quantification

Trend confidence ranges include two sources of uncertainty. First, we account for observational uncertainties in the CCFs, treating the two most uncertain CCFs, namely EIS and aerosol, separately from the rest (Appendix A6 and Figs. A3–A4). We thus perform our observational CCF analysis with all possible combinations of five EIS estimates, two aerosol estimates (CAMS or MERRA2), and two estimates for the remaining set of CCFs (ERA5 or MERRA2). This yields a 20-member ensemble of CCF sensitivities and thus radiative trend contributions from Eq. A1. We interpret this ensemble range as a measure of observational uncertainty for the cloud feedback, aerosol adjustment, and unforced variability components of the trend (Appendix A4). Second, for the GHG adjustment trend we take the spread in CMIP6 model-simulated GHG adjustment as a measure of uncertainty, using eight models with available data (Table A1).

For the total reconstructed trend, we then account for both uncertainties: we combine our eight estimates of the  $R_{\rm SWlow}$  295 GHG adjustment trend with the 20 estimates of the sum of other trend contributions (aerosol adjustment, cloud feedback and unforced variability), yielding a 160-member ensemble. For all trend components as well as the total reconstructed trend, our central estimate is the mean across ensemble members.

Uncertainty in the observed  $R_{\rm SWlow}$  trend is based on Raghuraman et al. (2021)'s uncertainty estimate of  $\pm 0.1~{\rm W\,m^{-2}\,K^{-1}}$  ( $\pm 1\sigma$ ) for CERES global-mean trends. We apply this to the global  $R_{\rm SW}$  trend due to all cloud types, and assume that low and non-low clouds contribute equally and independently to this uncertainty. This results in a  $\pm 1\sigma$  range of  $\pm 0.1/\sqrt{2} = \pm 0.07$  W m<sup>-2</sup> K<sup>-1</sup> for the observed global  $R_{\rm SWlow}$  trend.

## A6 Uncertainties in cloud-controlling factor trends

Reanalysis products suffer from known issues in their representation of long-term trends, introducing uncertainty in our analysis. To assess this uncertainty, in Fig. A3 we display the timeseries of near-global, standardised CCF anomalies. Fig. A4 shows the corresponding radiative contributions, obtained per Eq. A1.

The largest trend uncertainties are in the EIS and aerosol CCFs (Figs. A3b,g,h and A4b,g). For EIS, given the reanalysis uncertainty we include three datasets in addition to ERA5 and MERRA2: an EIS estimate based on the JRA3Q reanalysis product; and two satellite-based estimates of 700 hPa temperature,  $T_{700}$ , from AIRS and CLIMCAPS, combined with ERA5  $T_{\rm sfc}$ . (Note that EIS is based upon the difference between  $T_{700}$  and  $T_{\rm sfc}$ , and  $T_{\rm sfc}$  trends are well-constrained per Fig. A3a.) The five observational estimates of EIS disagree on the sign of the trend, ranging from  $-0.12~{\rm decade^{-1}}$  (AIRS) to  $+0.16~{\rm decade^{-1}}$  (ERA5). To provide additional context for these trends, we analyse an extended, five-member *amip* experiment with historical forcings up to 2014 and SSP2-4.5 from 2015 onwards, where sea-surface temperatures and sea ice are taken from HadISST1 (Rayner et al., 2003). The HadGEM3 simulations yield an ensemble-mean EIS trend of  $+0.04~{\rm decade^{-1}}$ , in the middle of the observational uncertainty range. While HadGEM3 may misrepresent aspects of the physics relevant to the EIS

**Figure A3.** Observational estimates of cloud-controlling factor (CCF) anomalies. Timeseries show standardised, deseasonalised monthly anomalies relative to the mean of the first 10 years, smoothed with a 12-month centred running mean; decadal trend values are shown in the top left corner of each panel. For each variable, the standardization is done using the ERA5 or CAMS standard deviation, so that the magnitudes are comparable across datasets. Values are near-global averages (60 °S to 60 °N). In (b), in addition to four observational estimates, we include values simulated by a five-member ensemble of extended *amip* simulations with the HadGEM3 climate model (not used in the CCF analysis). The HadGEM3 ensemble mean is shown in thick black, with grey shading denoting the ensemble range.

trend, the model's atmospheric state is known perfectly and hence there is no observational uncertainty. The HadGEM3 result thus provide some confidence that the "true" EIS trend lies somewhere in the range of observational uncertainty.

For the aerosol CCF, the two reanalysis datasets used here, CAMS and MERRA2, show very distinct time evolutions of sulfate AOD (Fig. A3g). While CAMS exhibits a relatively gradual AOD decrease, with a trend of  $-0.19 \,\mathrm{decade^{-1}}$ , MERRA2 shows a much more abrupt decline in the early 2010s, with a stronger linear trend of  $-0.37 \,\mathrm{decade^{-1}}$ . While we consider the CAMS time evolution to be more realistic, we nevertheless include the MERRA2 AOD data in our analysis to gauge the sensitivity of our results to the choice of aerosol dataset. The large differences in time evolution turn out to have only a very

**Figure A4.** As in Fig. A3, but showing the contributions of each CCF dataset to the near-global  $R_{\rm SWlow}$  anomalies, calculated relative to the mean of the first 10 years. Curves are averages over the corresponding ensemble members: for example, the dark blue curve in (a) is an average over estimates based on every possible combination of five EIS datasets, two log(AOD) datasets, and ERA5 data for the remaining five CCFs, yielding 10 ensemble members.

limited impact on our estimate of the aerosol-driven radiative trend (Fig. A4), possibly owing to compensating effects between sensitivity differences and CCF trend differences.

In addition to sulfate AOD, in Fig. A3h we also consider sulfate mass concentration,  $\log(s)$ , in the lower troposphere (925 hPa for CAMS, 910 hPa for MERRA2), following the work of Wall et al. (2022). Although the sensitivities resemble those obtained from the  $\log(\text{AOD})$  CCF (not shown), the CCF trends are in disagreement, with CAMS showing a weak decrease and MERRA2 a weak increase in mass concentration. This is contrary to our expectation of a clear decrease in sulfate aerosol concentration, particularly following the introduction of new shipping regulations in 2020. We therefore choose to employ  $\log(\text{AOD})$  as our sulfate aerosol CCF.

335

340

330 *Author contributions.* PC collected the data, performed the data analysis and wrote the initial draft of the paper. TA provided the HadGEM3 model data used in Fig. A3b. All authors reviewed the initial draft and contributed to the final version of the paper.

Competing interests. At least one of the (co-)authors is a member of the editorial board of Atmospheric Chemistry and Physics.

Acknowledgements. We are grateful to Senne van Loon for discussion of EIS trends, and to Omer Cohen and Guy Dagan for discussion of the choice of aerosol CCF, and to Jonathan Gregory for discussion of the energy imbalance trend. PC was supported by UK Research and Innovation (UKRI) under the UK government's Horizon Europe funding Guarantee (grant EP/Y036123/1). PC was additionally supported through UK Natural Environmental Research Council (NERC) grants NE/V012045/1 and NE/T006250/1. SWK and PN were supported by NERC grant NE/V012045/1. HA was supported by the Helmholtz Association through PoF IV in the Research Field Earth and Environment under the programme Changing Earth – Sustaining our Future. The effort of MZ was supported by the U.S. Department of Energy (DOE) Regional and Global Model Analysis program area and was performed under the auspices of the DOE by Lawrence Livermore National Laboratory under Contract DE-AC52-07NA27344. TA was supported by the Met Office Hadley Centre Climate Programme funded by DSIT. This work used JASMIN, the UK's collaborative data analysis environment (https://jasmin.ac.uk). We acknowledge the World Climate Research Programme's Working Group on Coupled Modelling, which is responsible for CMIP, and we thank the climate modelling groups for producing and making available their model output. We also thank the Earth System Grid Federation (ESGF) for archiving the model output and providing access, and we thank the multiple funding agencies who support CMIP and ESGF.

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
