# Peer review of "Emerging low-cloud feedback and adjustment in global satellite observations"

_EGUsphere, 2025_

## Author Comment (AC1)

**Response to reviewer comments**

We are grateful to both reviewers for their insightful comments and careful reading of our manuscript, and we are delighted that both reviewers thought the study was an important contribution to our understanding of global radiative trends.

We have made a number of changes following the reviewers' feedback. The most important of these are summarised below:

1) **Aerosol CCF trends.** A relevant analysis of global aerosol trends and their radiative impacts has been published since our original manuscript was submitted (Park and Soden 2025, doi: 10.1126/sciadv.adv9429). Park and Soden compare various observational and reanalysis datasets, and argue that Southern Hemispheric aerosol increases (driven by wildfires and volcanic emissions) largely compensate for Northern Hemispheric decreases, to yield a very small contribution to global-mean radiative trend.

This inter-hemispheric compensation is present in the aerosol reanalyses used here, but to varying degrees: sulphate aerosol optical depth (AOD) shows much less compensation than sulphate lower-tropospheric mass concentration (s) – see Fig. A2 in the manuscript. Correspondingly, global log(AOD) decreases globally during the study period (dominated by the Northern Hemispheric signal), whereas global log(s) trends are small and of either sign (Fig. A4). This dataset dependence is also discussed in Park and Soden (2025).

In Appendix 6 of the original submitted manuscript, we had reasoned that the log(s) trends were implausible, and used log(AOD) only as our aerosol CCF. In light of the new Park and Soden study, we now include sulphate mass concentration, log(s), as an additional aerosol CCF. With the two reanalyses CAMS and MERRA2, this yields a total of four aerosol datasets included in our study, providing a better estimate of observational uncertainty in aerosol effects.

Consistent with log(s) showing muted global trends, our revised estimate of the aerosol contribution to global low-cloud radiative trends is half of the original value, with a substantially increased ±1σ uncertainty (0.03±0.03 versus 0.06±0.01 W m$^{-2}$ decade$^{-1}$ previously). Other trend components are also affected: the cloud feedback contribution increases by 0.02 W m$^{-2}$ decade$^{-1}$; and the total forced and unforced contributions decrease by 0.02 W m$^{-2}$ decade$^{-1}$ each. The ±1σ uncertainty in the forced component increases by around 50%, consistent with the greater uncertainty in aerosol effects. (Uncertainty ranges are also slightly affected by changes made under point 2 below.) Changes in non-aerosol trend contributions are partly because the CCF sensitivities are inter-dependent: when the aerosol CCF is changed, the sensitivities to all other CCFs are also affected, because the regression model partitions the variance differently across CCFs.

We note that our estimate of the aerosol-cloud radiative trend contribution (0.03 ± 0.03 W m$^{-2}$ decade$^{-1}$) is in excellent agreement with the numbers of Park and Soden (2025) when compared like-for-like. Using sulphate mass concentration from MODIS and MERRA2 they obtain trends around 0.00 W m$^{-2}$ decade$^{-1}$, while with CAMS sulphate mass concentration they find 0.06 W m$^{-2}$ decade$^{-1}$.

The total reconstructed low-cloud radiative trend is now 0.17 W m$^{-2}$ decade$^{-1}$, underestimating the real observed trend by 0.05 W m$^{-2}$ decade$^{-1}$ (instead of just 0.01 W m$^{-2}$ decade$^{-1}$ previously). We find this underestimate to be acceptable considering uncertainties in the methodology, the CCF trends, and the CERES observed radiative trend itself, and we note that the ±1σ uncertainties in the observed and reconstructed trends strongly overlap (Fig. 1d). The discussion in section 3 has been revised to reflect the new numbers and the larger unexplained residual.

**2) Uncertainty calculation.** We have revised our trend uncertainty calculation to account for uncertainty in the decomposition between forced and unforced components of the CCF trends (and hence the components of the SWCRE trend). As a reminder, we use the CMIP6-mean CCF trends during 2003–2024 as our best (if imperfect) estimate of the forced trends. We then use this estimate to decompose trends into a forced, SST-mediated component, and an unforced residual (with the contribution of GHG adjustments separately accounted for; Eqs. A4–A5).

The uncertainty in this decomposition is now calculated from CMIP6 inter-model spread in the 2003–2024 forced CCF trends. Ideally, such spread would be obtained from multiple large ensembles from each CMIP6 model – where each model's ensemble-mean trend approximates the model's true forced response. Since we have only one realisation per model, we instead use bootstrapping: we generate 1000 synthetic 30-member ensembles of the 2003–2024 CCF trends and thus SWCRE trends, calculate the bootstrapped ensemble means, and use the spread across these ensemble means as our uncertainty estimate. This uncertainty applies to both the SST-mediated (i.e. "cloud feedback") trend component, and the unforced trend component, since the latter is calculated from the former. Note that this uncertainty is in addition to (and assumed independent of) the two already included uncertainty sources – namely observational uncertainty in the CCF trends, and the CMIP6 spread in low-cloud GHG adjustment. The full uncertainty calculation is explained in Appendix A5, where the text has been substantially revised.

The inclusion of this additional uncertainty has an overall minor impact on the ranges: the uncertainty range for the low-cloud feedback trend increases by less than 0.01 W m$^{-2}$ decade$^{-1}$; the ranges for the forced component and the total reconstructed trend increase by less than 0.005 W m$^{-2}$ decade$^{-1}$.

**3) Nomenclature.** As a more minor change, in response to Reviewer 2's suggestion we now denote low-cloud radiative anomalies as $SWCRE_{low}$, instead of $R_{SWlow}$ previously. This better reflects that the observed radiative fluxes are calculated as cloud-radiative effects from CERES-FBCT data. The possible impact of cloud masking effects on $SWCRE_{low}$ trends is discussed in greater detail in our replies to reviewer 2's comments.

**Reviewer 1**

Ceppi et al. use cloud-controlling factor (CCF) analysis to investigate the role of low clouds in the observed trends of Earth's absorbed solar radiation. They combine the CERES-FBCT satellite product with reanalysis and model data to quantify how CCFs influence low clouds and how those in turn affect the amount of absorbed solar radiation. They decompose the trend into unforced and forced components, where the latter is further split into cloud feedback (to surface warming) and rapid aerosol and GHG adjustments. They find that all three of them contribute to a forced radiative trend, with cloud feedback being the strongest contributor, whereas the unforced component seems smaller but very uncertain. In my view the study is a well-founded, important and timely contribution to the dynamically evolving understanding of the observed radiative trends and certainly suitable for publication as an ACP letter, although I think that some points would benefit from clarification.

We thank the reviewer for their positive assessment and thoughtful comments.

Before providing my specific comments, I'd like to mention that I'm wondering how these results based on local CCFs fit (or not) with recent work that links the observed cloud and SW absorption trends primarily to large-scale circulation trends rather than local "within-regime" factors (primarily Tselioudis et al. 2025). Could the authors add a brief discussion about this?

Dynamically-driven shifts in cloud regimes (for example due to storm-track shifts as discussed in Tselioudis et al. 2025) would be captured by the CCF method via the induced CCF anomalies: for example, storm-track shifts would affect cloud-radiative properties via their impact on surface wind speed (WS), SST advection (SSTadv), vertical velocity ($\omega_{700}$), etc. A few studies have used the CCF approach to quantify the impact of midlatitude storm-track shifts on low clouds, e.g. Zelinka et al. 2018 (10.1175/JCLI-D-18-0114.1) and Grise and Kelleher 2021 (10.1175/JCLI-D-20-0986.1).

As a brief discussion, we have added the following text at the end of Appendix A3: "Previous studies have demonstrated the ability of the CCF analysis method to capture cloud-radiative anomalies, whether driven thermodynamically (e.g., the response to global warming; Myers et al., 2021; Ceppi et al., 2024) or dynamically (e.g., storm track shifts; Zelinka et al., 2018; Grise and Kelleher, 2021)". (Note that while we do not cite Tselioudis et al. 2025 here, it is cited elsewhere in the paper.)

Overall, I recommend to publish this paper as an ACP letter subject to minor revisions.

Specific comments (including minor technical ones):

L5: "The contribution of natural climate variability is weak but uncertain [...], owing to a poorly constrained trend in boundary-layer inversion strength"; It remains somewhat unclear to me why the large uncertainty related to trends in EIS should affect the unforced component much more than the forced component. Can this be clarified?
This is because the forced component of the response includes only relatively weak EIS anomalies, and most of the EIS signal is therefore attributed mainly to the unforced component of the response (see Fig. A2, 2nd row). We have added a sentence to explain this where the uncertainty in the unforced component of the trend is discussed, at the end of section 3 (L94–96): "This EIS trend uncertainty also dominates the spread in the total reconstructed SWCRE$_{low}$ trend (r = 0.63; Figs. A4–A5), *while only making a limited contribution to uncertainty in the forced component of the trend, owing to a weak forced EIS response (Fig. A2)*" (new text in italics).

L9 (and elsewhere): "... processes other than low clouds."; maybe this is a bit meticulous, but "low clouds" are not a process. Maybe reformulate to something like "... processes unrelated to low clouds." or so.
Fixed, thanks.

L37: "SW low-cloud anomalies [...] made a large contribution amounting to half of this decadal trend [of Earth's global energy imbalance]"; It seems that the CERES total SW trend is even something like 0.8W/m2/dec, counteracted by a LW trend of around -0.3W/m2/dec (e.g., Fig. 3 in Myhre et al. 2025). Do I infer correctly that (i) the contribution of low clouds to the total SW trend is only around a quarter and that (ii) the bulk of the rest would then likely be due to mid- and high-level clouds (with more of a LW compensation)? Maybe that's worth to mention.

Agreed – and the reviewer is correct that low clouds account for about a quarter of the total trend in absorbed solar radiation (ASR) of 0.86 W m$^{-2}$ decade$^{-1}$. Given that the SWCRE trend (from all clouds) is 0.43 W m$^{-2}$ decade$^{-1}$, i.e. only half of the ASR trend, it is however not the case that the remainder of the trend is mainly attributable to non-low clouds. This means other changes – surface albedo, water vapour absorption, or shortwave forcing from insolation or aerosols – must also be playing some role. We have added the following text at the end of the first paragraph of section 2, and note that for brevity we do not discuss the SWCRE trend: "Low clouds however only account for about a quarter of the trend in absorbed solar radiation (0.86 W m$^{-2}$ decade$^{-1}$, not shown; Loeb et al., 2024b; Myhre et al., 2025), which includes additional contributions from non-low clouds, surface albedo, water vapour absorption, and shortwave forcing."

Figure 1: (i) I recommend to add units to the decadal trend numbers. (ii) In panel d, it would be good to mention/remind that this is about LOW-CLOUD trends, e.g., by changing the title into "Low-cloud trend contributions". (Also in the corresponding part of the caption.)

We have added units to each panel – and moved the trend numbers the lower-right corner of each panel, owing to space issues. We have also changed the title of panel d accordingly, and edited the caption.

L50: "... nearly all of the RSWlow trend is associated with decreasing cloud amount, as opposed to decreasing optical depth"; I'm wondering if observational uncertainties in the way how cloud amount and optical thickness are distinguished (including the choice of a threshold from which point something is considered cloud vs. no cloud) might affect this. Related, the near-global averaged C_low seems to have much less of a trend component than R_SWlow, and in particular does not mirror the increase of R_SWlow of the last few years, making me wonder if this could be related to recent aerosol-induced changes in cloud optical thickness after all.

We agree with the reviewer's point. There are however other reasons why the trend in $C_{low}$ may not perfectly match the radiative trend; for example, the radiative impact of a change in cloud fraction will depend on the seasonality and geographical location of the clouds – the same trend in $C_{low}$ will have a greater effect when insolation is stronger. We are constrained by space limits, but we have added a note that the partitioning between cloud amount and optical depth contributions is subject to observational uncertainty (L57).

Fig. 2: I recommend to add units to the decadal trend numbers. (This also holds for Figs. A3 and A4.)

We have made this change for Fig. 2, consistent with Fig. 1. For Figs. A3 and A4 (now numbered A4 and A5), given the large number of trend values provided, and the fact that the units are consistent for each figure, rather than repeating the units many times we now specify them in the figure captions.

Fig. 3: Given that GHG adjustment seems to be an important contributor, I'd like to see a version of panel f with zoomed colorbar, plus a minimum explanation/hypothesis in the text indicating the possible physics behind this adjustment.

This has been added as an extra appendix figure (new Fig. A3). We have also added a sentence explaining the basic physics of this positive adjustment – namely lower-tropospheric warming and drying, plus a reduction in cloud-top radiative cooling in an optically thicker atmosphere, with suitable references (L84–86).

L81: "... given the known large decadal variations in low-cloud feedback."; is this indeed meant in the sense of decadal variations that would change (temporarily) the "background state" which would then result in modified low-cloud FEEDBACK during that limited time period? Or does this actually not relate to the feedback but just the clouds themselves, so something like "... given the known large decadal variations in low-cloud cover.", which I consider much more plausible? Or, a third option, is it about decadal variations in DIAGNOSED low-cloud feedback, given that limited observational periods will certainly affect estimates of the feedback?

*We mean the first option: the driver of the changes in cloud feedback is the time variation in the pattern of SST warming (see the literature cited in the manuscript). To make this less ambiguous, we have rephrased this as "… the known large decadal variations in low-cloud feedback associated with time-varying SST patterns" (new text in italics).*

Paragraph starting L80: Is it possible that the smallness of the uncertainty in the forced R_SWlow component is partly due to the assumption that the GMST trend is completely forced?

*We no longer make this assumption – please see point 2 in the summary of our main changes on page 1 of this document. Furthermore, as a result of other methodological changes the uncertainty in the forced component has increased. We have therefore removed the sentence stating that the uncertainty of the forced component was relatively small.*

L97: "Comparing with amip minimises differences in CCF trends between models and observations, thus highlighting the role of the cloud-radiative sensitivities."; Is "minimises" here really the case? I mean, aspects like EIS, which as you show are more related to T_700hPa than T_srf, may still be rather unconstrained by the prescribed SST. Maybe "reduces" would be more appropriate?

*We have changed to "reduces". Based on previous literature, free-tropospheric changes (in EIS and other variables) should be relatively well constrained by the SST boundary conditions, though of course not perfectly.*

L138: "the observed substantial low-cloud radiative trend cannot be interpreted as evidence of an unexpectedly strong low-cloud feedback that climate models are systematically missing (Goessling et al., 2025)"; The Goessling et al. paper does not make a strong statement about this being the main culprit, but mentions an upper-range low-cloud feedback just as one of three possible contributors (besides aerosols and variability).

*We agree that Goessling et al. did not imply this (and we did not mean to imply that they did!), so we have removed the reference here to avoid confusion.*

Paragraph starting L165: In principle one can retrace the CCFs to earlier literature where there's more explanation and physical argumentation around them. However, given that this chain of studies is somewhat long/complex, I would find it helpful if brief explanations of the physical rationale (and definition, see next point) of these seven CCFs could be repeated here.

*We agree with the reviewer that a summary of the physical basis for the CCFs would be helpful. The review paper of Klein et al. 2017 (10.1007/s10712-017-9433-3) provided a synthesis of our physical understanding of low-cloud CCFs in their Table 1, along with references to relevant studies. Since we are already at the length limit, rather than paraphrasing this text here we point the reader to this review: "The physical relevance of the six meteorological controlling factors is reviewed in Table 1 of Klein et al. (2017)" (L183).*

L167: "sea-surface temperature (SST) advection, SSTadv"; if I'm not mistaken, this is about temperature advection by near-surface winds, where the SST gradient strongly governs the air-temperature gradient, but it's not the same as actual "sea-surface temperature advection", which sounds like ocean surface velocities were involved. Some clarification would be good.

*The reviewer's understanding is correct, and we agree that the wording was ambiguous. We have rephrased as "horizontal air-temperature advection across the SST gradient".*

L203: Does this equation need to be applied iteratively from top (low pressure) to bottom (high pressure), so that U(p_i) is then always something like sum(L_n(p<p_i))? And does that ultimately yield a total cloud cover that is consistent with the CERES total cloud cover, or am I thinking wrong here?

In the equation, reproduced below for clarity,

$$L_n(p,\tau) = \frac{L(p,\tau)}{1-U}$$
,

U is not a function of pressure level p, and represents vertically-integrated upper-level cloud amount (integrated over all non-low pressure bins). Therefore $L_n(p,\tau)$ is independent of what happens in other low-cloud bins, and the calculation can be performed in any sequence of p and τ bins. We have added a sentence to clarify that U is independent of p (L225).

On the reviewer's second point: by design, the calculation produces a greater low-cloud cover than what is reported by passive satellite retrievals (such as the MODIS retrievals used for CERES-FBCT), because we are "unmasking" low clouds that would otherwise be masked by upper-level clouds. However, the resulting radiative anomalies are rescaled by the climatological upper-level cloud-free area $(1 - \bar{U})$:

$$\mathrm{SWCRE_{low}} = (1-\bar{U})\sum_{p=1}^{2}\sum_{\tau=1}^{6} K_{\mathrm{SW}}\, L_n$$

Dividing by $(1 - U)$ (the *instantaneous* upper-level cloud-free area) and then re-multiplying by $(1 - \bar{U})$ (the *climatological* upper-level cloud-free area) ensures a correct scaling of the low-cloud radiative effects, while also removing the impact of variations in upper-level cloud $U$ on top-of-atmosphere radiative fluxes. This "cloud obscuration" effect and its radiative implications are discussed in greater depth in Zelinka et al. 2025 (10.5194/acp-25-1477-2025).

L235: "The sensitivities Θi are calculated via ridge regression, where all variables have been deseasonalised, and the CCF predictors have been standardised"; I do not quite see the justification of computing "all-year" sensitivities given that they could well vary considerably seasonally (maybe as much as regionally in places?) due to seasonal changes of the background conditions, in particular in the extratropics. Is there evidence that this is not the case? I think that should be clarified.

We agree with the reviewer in principle, and one could calculate separate sensitivities for different seasons. In practice, given the limited length of available data, it would be difficult to accurately assess seasonal differences in the sensitivities. Furthermore, the method used here was shown to have excellent out-of-sample predictive skill for low-cloud feedback in a perfect-model setting, and also worked well in predicting observed extreme anomalies out-of-sample (Ceppi et al. 2024, 10.1029/2024GL1105250). We expect this to be true wherever we predict multi-year (as opposed to monthly or seasonal) anomalies in cloud-radiative effects, as is the case here.

To clarify this point, we have added the following text (L264): "Following prior studies, we ignore any seasonality or mean-state dependence of the sensitivities Θi, which have been shown to have strong out-of-sample predictive skill in both models and observations (Ceppi et al., 2024)."

L265: Given that dX/dT_for and dX/dT_GHG are based on different sets of models, I'm wondering if dX/dT_for and thus the resulting dX/dT_SST would be similar if just the same subset of models was used?

Using a larger set of models seems preferable to us in order to minimise the impact of unforced natural variability, although we acknowledge that the use of a different set of models for the GHG adjustment calculations is not ideal. In any case, we have repeated the calculation of the forced CCF response using the smaller set of 8 models that provide piClim simulations, and the results are reassuringly robust. Fig. R1 below shows the CCF trend decomposition based on the reduced set of models (for comparison with Fig. A2 in the manuscript, based on the full set of models): the forced SST pattern is qualitatively similar, and the forced responses for other CCFs remain small

compared to the unforced responses. In terms of the radiative impacts, changes are likewise minimal: with the reduced set of models, the estimated forced trend contribution decreases by 0.01 W m⁻² decade⁻¹, and correspondingly the unforced contribution increases by the same amount (since it is calculated as a residual).

We have added the following discussion at L296: "Although different sets of CMIP6 models are used for the calculation of $dX/dT_{for}$ and $dX/dT_{GHG}$ (Table 1), we obtain a very similar decomposition of the CCF trends if we instead use a smaller common set of eight models to calculate both $dX/dT_{for}$ and $dX/dT_{GHG}$ (not shown)".

[Figure]

*Fig. R1. As in Fig. A2 of the manuscript, but the partitioning between forced and unforced CCF trends is based on a reduced set of 8 models that also provide piClim simulation data for the calculation of the GHG CCF adjustments.*

L266: Similarly, here dX/dT (obs-based) and dX/dT_for (model-based) stem from different datasets, so I'm wondering how the resulting dX/dT_unfor would look like if they were obtained from consistent data. For example, if one would use a single ensemble member of a CMIP historical/scenario model simulation as surrogate observation and base dX/dT_for on a (large) ensemble of just that same model, would the diagnosed unforced components exhibit quasi-random patterns of similar magnitude (compared to the right column of plots in Fig. A2)? Could that provide evidence for the validity of the method, whereas, if magnitudes are much smaller, would that suggest that the "unforced" components found here may contain considerable amounts of in reality forced changes?

We agree that analysing large ensembles would be an ideal way to assess the forced and unforced components of the response, and their inter-model uncertainty. We are unfortunately unable to test the reviewer's idea because we do not have data from large historical/SSP ensembles. Many CMIP6 models only provide a limited number of historical realisations; and where relatively large ensembles do exist (e.g. the UK Met Office models), we lack the resources to retrieve and process the large data volumes involved, considering we require a substantial number of variables for our analysis. There are many open questions regarding the ability of CMIP models to represent the "true" pattern of forced response to anthropogenic forcing, so we caution readers that our decomposition between forced and unforced responses needs to be interpreted with caution; see the text at L310–316.

L293+294: "for the GHG adjustment trend we take the spread in CMIP6 model-simulated GHG adjustment as a measure of uncertainty, using eight models with available data" and "we combine our eight estimates of the RSWlow GHG adjustment trend with the 20 estimates of the sum of other trend contributions [...], yielding a 160-member ensemble"; my understanding is that the GHG adjustment is derived from the piClim-ghg/control simulations, and according to Tab. A1, that data is available from ten models, not eight. Have I misunderstood this?

There was an error in Table A1, apologies. Contrary to what was indicated, MIROC-ES2L and MIROC6 do not provide ISCCP simulator output for piClim-ghg, and are therefore not included in the calculation of the GHG adjustment. This is now fixed.

L326: "the CCF trends are in disagreement, with CAMS showing a weak decrease and MERRA2 a weak increase in mass concentration. This is contrary to our expectation of a clear decrease in sulfate aerosol concentration, particularly following the introduction of new shipping regulations in 2020"; Could this also be related to natural variations in aerosol concentrations (e.g., wildfires, even if sulfate is not a typical wildfire aerosol)?

Yes – please see our summary of main changes on p1, and our replies to reviewer 2's first major comment. Given the recent study by Park and Soden (2025), we now treat the various aerosol products as equally plausible representations of real-world aerosol trends. Correspondingly, the text in this section has been reworked.

References:

Tselioudis et al. (2025), Contraction of the World's Storm-Cloud Zones the Primary Contributor to the 21st Century Increase in the Earth's Sunlight Absorption, https://doi.org/10.1029/2025GL114882

Myhre et al. (2025), Observed trend in Earth energy imbalance may provide a constraint for low climate sensitivity models, https://doi.org/10.1126/science.adt0647

**Reviewer 2**

This paper uses satellite observations, reanalysis, and climate model data in a cloud controlling factor (CCF) analysis framework to estimate the contribution of low cloud changes to the shortwave (SW) component of the trend in Earth's energy imbalance (EEI) for July 2003 to June 2024. They find that low cloud changes alone contribute 0.22 Wm-2 per decade to the trend in SW top-of-atmosphere (TOA) radiation, which is substantial compared to the observed EEI trend of 0.44 Wm-2 per decade. Unfortunately, the paper does not state what the compensating LW contribution from low cloud changes is. The authors further show that the low cloud SW trend is a result of low cloud feedback, sulfate aerosol adjustment, and GHG adjustment, which account for 82% of the SW low-cloud trend. The authors claim that natural climate variability plays a minor role in explaining the trend. Comparisons between observation-based results and CMIP6 AMIP and historical global climate simulations suggests that the observed SW low-cloud trends lie within the range of the simulated trends. The implication is that underestimation of the observed EEI trend by state-of-the-art climate models reported in earlier studies is unlikely to be due to model representation of low cloud changes.

This is a very interesting paper that makes an important contribution to our understanding of the trend in EEI. I only have two major comments and numerous minor ones.

We thank the reviewer for their thoughtful comments on the manuscript, and are pleased that they found the paper interesting and useful.

Major Comments:

Considering recent work (likely) published after this manuscript was submitted, it would be helpful if the authors commented on the work by Park and Soden (2025; https://www.science.org/doi/10.1126/sciadv.adv9429), who did a very similar analysis but reached a very different conclusion about the role of aerosols in explaining the SW low-cloud trend. Park and Soden (2025) used many of the same datasets and analysis steps as in the present paper. While the present paper finds that aerosols contribute 0.06+/-0.01 Wm-2 per decade (29%) to the 0.21 Wm-2 per decade low-cloud SW trend, Park and Soden (2025) find the aerosol contribution is negligible (–0.006+/-0.028 W m–2 per decade) primarily due to compensation between decreases in aerosol concentration in the northern hemisphere and increases in the southern hemisphere. Importantly, the difference between estimates from these two studies exceeds the stated uncertainties. A key methodological difference is that Park and Soden (2025) use the natural logarithm of sulfate mass concentration at 925 hPa (SO4) from MERRA-2 & CAMS, the natural logarithm of MODIS Aerosol Index, and cloud droplet number concentration from MODIS while the present study uses MERRA-2 AOD in the CCF regression. Park and Soden (2025) considered two approaches: that of Wall et al (2022) and a scheme that explicitly accounts for aerosol activation rate when determining susceptibility of the SW low-cloud radiative effect to variations in aerosol concentration. Considering the discrepancies between the two studies, it seems worthwhile for the authors to include a comment about the Park and Soden (2025) paper and perhaps revise their apparent confidence in their aerosol result. A further complicating factor is that CMIP6 climate models simulations suggest an even larger contribution by aerosol-cloud interactions (Hodnebrog et al., 2024), but those are questionable given that the aerosol forcing data used are outdated.

We agree; please see the text on p1, where we describe the main changes to our manuscript. In summary, we now include sulphate lower-tropospheric mass concentration, log(s), from both CAMS and MERRA2 as additional aerosol CCFs – so that we use four instead of two aerosol CCFs in total. This has two impacts on our results: first, the central estimate for the aerosol contribution to the $R_{SWlow}$ trend is weaker, 0.03 instead of 0.06 W m-2 decade-1; second, the uncertainty in the aerosol contribution is greater. This probably better reflects the true observational and methodological uncertainty in this estimate. Our numbers are also in good agreement with the estimates of Park and Soden (2025).

An additional impact is that the total reconstructed trend (grey lines in Fig. 1b) is now weaker, and underestimates the actual observed trend by 0.05 W m$^{-2}$ decade$^{-1}$. We find this discrepancy to be acceptable considering the uncertainties in the CCF reconstruction and in the observed trend itself; this is reflected in the overlapping uncertainty ranges in Fig. 1d (black and grey bars).

The reviewer's final point on the potentially unrealistic representation of historical aerosol concentrations in CMIP6 simulations is well taken. However, such an error should not directly affect our findings. For the comparison between CERES observations and AMIP simulations in Fig. 1c, the AMIP trend is inferred from emulated timeseries based on reanalysis (rather than simulated) aerosol fields – the same four datasets as used for the reconstruction of observations. As for the results in Fig. 4, based on historical simulations, they are based on the period 1995–2014 where the aerosol concentration trend is weak in CMIP6 models (with a rapid decrease from 2015 onwards in the SSP2-4.5 scenario).

The authors state that low cloud changes make a substantial contribution to the trend in EEI but only quantify the SW component. For completeness, they should also quantify the compensating LW trend contribution from low cloud changes. If that is not feasible, instead of comparing the SW low cloud contribution to the EEI trend (i.e., 0.44 Wm-2 per decade), they should compare its contribution to the total trend in SW, which is much larger (~0.8 Wm-2 per decade).

The LWCRE contribution can be calculated from CERES-FBCT data, but its interpretation is difficult because the trend contains a large cloud masking component – i.e. effects due to non-cloud factors (mainly water vapour, temperature, greenhouse gas forcing) affecting the difference between all-sky and clear-sky radiation and thus the LWCRE. This issue is discussed in Appendix A1, L165–173. Therefore rather than discussing LWCRE trends, we have added text in section 2 where we compare our numbers with the total trend in absorbed SW of 0.86 W m$^{-2}$ decade$^{-1}$ for the chosen period. Specifically, we have added the following text (L42):
"Low clouds however only account for about a quarter of the large increase in absorbed solar radiation (0.86 W m$^{-2}$ decade$^{-1}$, not shown; Loeb et al., 2024b; Myhre et al., 2025), which includes additional contributions from non-low clouds, surface albedo, water vapour absorption, and shortwave forcing."

Minor Comments:

Line 28-33:
Why start in July 2003? The combined Terra and Aqua data are available since July 2002, and Terra-only since March 2000.
Apologies that this wasn't discussed. CERES-FBCT is available from July 2002, but the CAMS aerosol reanalysis is only available from January 2003. For practical reasons, we perform the analysis on the most recent full 21-year record, July 2003 to June 2024. This is now noted in the first paragraph of the Data section (Appendix A1). We also point the reader there at the very start of section 2 (L36).
Note that climate model forcing and adjustments are also used alongside observations to diagnose low cloud radiative trends.
We have added a mention here that CMIP6 model simulations are used to complement CERES-FBCT observations.

Line 37: Please note that SW is defined positive downward.
We have added "defined positive down".

Line 39: "…amounting to half of this decadal trend".
What about LW? Is the trend contribution zero? The text makes it seem like low cloud changes alone account for 50% of the trend in Earth's energy imbalance. That is only true if their contribution to the LW trend component is zero, which is assumed but not shown.

The LW impact of low-cloud changes is generally constrained to be weak, because such clouds are at altitudes lower than the effective emission level. Consistent with this, previous assessments of low-cloud feedback suggest that the LW component is typically of opposite sign but an order of magnitude smaller than the SW component (e.g. Zelinka et al. 2016, 10.1002/2016GL069917). We see no reason that a similar scaling should not hold for the trends. Note however that this reasoning applies to *cloud-induced radiative anomalies*; LWCRE trends will be substantially larger in magnitude (and more negative in sign) because of non-cloud effects impacting CRE, as discussed above.

We have added the following sentence to Appendix A1 (L171): "We do not analyse LWCRE data here, since their trend is dominated by cloud masking effects (not shown), and furthermore low-cloud properties only have a small impact on top-of-atmosphere LW fluxes."

Line 43: "…low clouds exhibit a positive radiative sensitivity to surface temperature…"
This is unclear/vague. Please explain that "positive radiative sensitivity" implies increased absorption of solar radiation.
For clarity, we have rephrased this as "increasing surface temperature generally promotes less low cloud and thus anomalously positive $SWCRE_{low}$, whereas increasing estimated inversion strength has the opposite effect…", which we hope is clearer.

Figure 1d:
It seems the trends in Fig. 1d are derived from 12-month running mean time series, which exhibit substantial autocorrelation. However, there is no mention of how the trend uncertainties are calculated. Do the uncertainties account for autocorrelation in the data?
All trends are calculated from the raw monthly data, before the running mean is applied purely for visualisation purposes. This is now specified in the captions of Figs. 1 and 2.
It seems odd to include the cloud feedback contribution in the "Forced" category (second bar from bottom of Fig. 1d). Isn't it more appropriate for this to be a separate category (i.e., SST-mediated response to forcing?)
We see the reviewer's logic. However, the SST-mediated response is ultimately driven by the effective forcing (instantaneous plus rapid adjustments). Our preference is to emphasise the distinction between natural unforced variability, versus forced response – including the SST-mediated response to the forcing. The breakdown in Fig. 1d still allows readers to separate out the SST-mediated component from the other forced components (namely the adjustments to GHG and aerosols).

Lines 87-88: In light of the recent paper by Park and Soden (2025), who also used MERRA-2 and CAMS but considered the natural logarithm of sulphate mass concentration at 925 hPa as opposed to log(AOD), and found a negligible trend contribution from aerosols, is this an accurate statement. It seems that the aerosol contribution is highly uncertain.
With the addition of sulphate aerosol mass concentration as a CCF, the uncertainty in the aerosol contribution (and thus the total forced component) has increased (Fig. 1d). We have therefore removed this sentence.

Lines 150-151: "We use gridded global satellite observations of cloud amount and top-of-atmosphere radiative fluxes from the CERES Flux-By-Cloud-Type (CERES-FBCT) product (Sun et al., 2022)".
CERES-FBCT is available daily and monthly. Which of these is used here?
We now note here that this is monthly data (and this was already noted at the end of section A1).

Line 155: "Note that CERES-FBCT provides cloud-radiative effect rather than true cloud-induced radiative anomalies"
FBCT provides all-sky and clear-sky TOA fluxes along with TOA fluxes for 42 pc-tau cloud types along with their amounts in each gridbox. The all-sky flux is thus weighted sum of clear-sky and individual cloud type fluxes, where the weights are given by the respective clear or cloudy sky

amounts. Cloud-radiative effect can be computed from all-sky and clear-sky, but it's not part of the data product.

Fair enough – apologies for the imprecise wording. Our point, however, was that from the variables provided in CERES-FBCT we can calculate cloud-radiative effect, but not true cloud-induced radiative anomalies. We have therefore rephrased as, "Note that from CERES-FBCT, we can calculate cloud-radiative effect rather than true cloud-induced radiative anomalies…"

Lines 156-158: "These are however expected to be much smaller for SW than longwave (LW) fluxes, particularly since we exclude regions poleward of 60° where surface albedo changes are largest (Raghuraman et al., 2023)."

While this may be true for monthly anomalies, it isn't so obvious when considering trends, as is done here. For example, we know that there is a strong trend in water vapor, which affects SW TOA flux.

We are unable to calculate SW cloud masking trends for low clouds in isolation, but Raghuraman et al. 2023 (10.1175/JCLI-D-22-0555.1) provided numbers for the SWCRE masking trends for all clouds combined, for the period 2001–2020 (thus not too dissimilar from our study period). The main SW cloud masking effect was a negative contribution surface albedo, but this mainly originates from polar regions, which are excluded here. Water vapour effects made a small positive contribution of around +0.02–0.03 W m$^{-2}$ decade$^{-1}$ (eyeballing from Fig. 5 of Raghuraman et al. 2023). This suggests that the overall cloud masking effect should be relatively modest, which we believe is consistent with the wording used here.

Line 159: Please clarify what RSWlow is. The prior sentences discuss CRE, but Line 159 mentions low cloud SW anomalies. Is RSWlow the low cloud flux weighted by low cloud amount or is it CRE for low clouds? If it's more like CRE, consider using "CRE" instead of RSW in the name.

Throughout the paper, we have changed "$R_{SWlow}$" to "$SWCRE_{low}$" for clarity. This is a more accurate description of the fluxes calculated for CERES-FBCT. Note however that for the CMIP6 analysis, the radiative anomalies are computed from ISCCP simulator output combined with cloud-radiative kernels, and therefore constitute true cloud-induced radiative anomalies (i.e. without any cloud masking effects), rather than cloud-radiative effect anomalies. This is now clarified in Appendix A1, L203: "This calculation [of $SWCRE_{low}$ for CMIP6 models] isolates the radiative impact of cloud properties from other, non-cloud factors, and strictly speaking provides cloud-induced radiative anomalies rather than CRE."

Lines 162-163: "The cloud-radiative changes diagnosed from CERES-FBCT provide a close match to those obtained from CERES-EBAF (Loeb et al., 2024b)"

It is worth pointing out that this statement is true only for the Terra+Aqua period. It has not been demonstrated for the NOAA-20 period, which is considered in this study.

Also, FBCT is a daytime only product whereas EBAF accounts for both daytime and nighttime LW.

This is a fair point, and now noted in the text (L176): "The cloud-radiative changes diagnosed from CERES-FBCT provide a close match to those obtained from CERES-EBAF, *at least over the period covered by the Terra and Aqua satellites (Loeb et al., 2024b)*" (new text in italics).

Line 195: "CERES-FBCT data consists of clear-sky radiative flux Rclr, cloudy-sky radiative flux Rcld, and cloud amount."

It's worth pointing out that all-sky TOA flux in a gridbox is the sum of Rclr plus the sum of f(p, tau)*Rcld(p, tau) for all (p, tau) cloud types, where f(p, tau) is the cloud amount for cloud type (p, tau). Thus, the contribution to all-sky from a given cloud type is Rcld(p, tau)*f(p, tau).

Thanks – we have added text here and an equation (new Eq. A1) to explain this. Note that the all-sky TOA flux is $(1-f_{tot})*R_{clr} + \Sigma_p\Sigma_\tau f(p,\tau)*R_{cld}(p,\tau)$ – the $(1-f_{tot})$ scaling of $R_{clr}$ was missing from the reviewer's comment.

Lines 196-198: "Following previous literature, we categorise clouds in the lowest two bins (p > 680 hPa) as low clouds. We use only the shortwave (SW) component of the fluxes, denoted RSWclr and RSWcld."

This is confusing. How does RSWcld differ from RSWlow, introduced in line 159? It seems RSWcld is the cloudy SW radiative flux for a cloud in a given (p, tau) bin from the FBCT product while RSWlow is a derived quantity (see line 210) analogous to cloud radiative effect for a specific (p, tau) cloud type. Using such similar symbols (RSWcld and RSWlow) to define these quantities is unnecessarily confusing. Consider using something like "CRE_SW_low" instead of RSWlow to make the distinction clearer.

As discussed in reply to the previous point at L159, we now use "$SWCRE_{low}$" throughout, instead of the previous $R_{SWlow}$. This hopefully helps avoid confusion with the new terms $R_{SWcld}$ and $R_{SWclr}$ introduced here. The reviewer's interpretation of these two variables is correct.

Also, is RSWcld = cloudy-sky radiative flux (Rcld) times cloud amount? Please clarify.

No, this is just cloudy-sky (SW) radiative flux, by analogy with $R_{cld}$. To ensure this is clear, we now specify at L219 that "we use only the SW component of the *clear- and cloudy-sky* fluxes, denoted $R_{SWclr}$ and $R_{SWcld}$" (new text in italics).

Line 203: Equation for non-obscured low cloud amount Ln. A more straightforward equation for this is: Ln(p, tau) = L(p, tau) / (L+Clr), where L is total low cloud amount and Clr is the clear-sky amount.

We see the reviewer's logic, but the proposed notation doesn't make it clear that it is the *upper-level* clear-sky fraction in the denominator (and not the total clear-sky fraction). However, for clarity we have added the following text right after the equation: "…where 1–U is the upper-level clear-sky fraction".

Line 205: "absence of clouds"
Should "clouds" be preceded by "low" here?

This statement applies to any (p, tau) bin, not just for low clouds. This is consistent with the text: "…[this] quantifies how top-of-atmosphere radiation changes in the presence versus absence of clouds for each month and (p, tau) bin".

Line 206: "We can then calculate the low-cloud contribution to top-of-atmosphere radiative flux"
This sentence contradicts the sentence on line 211, which (correctly) describes the equation on line 209 as contribution of low clouds to SW cloud-radiative effect. Please clarify.

We have rephrased this as "We can then calculate the low-cloud contribution to top-of-atmosphere SWCRE, $SWCRE_{low}$…"

Line 209:
(a) Since the reader does not know if daily or monthly FBCT data are used, it's unclear if this equation refers to a monthly mean in a gridbox derived from daily FBCT values or a monthly or longer average of multiple gridboxes. Also, U overbar is not defined.

We have clarified that we are using monthly FBCT data (at the start of Appendix A1), and furthermore we now specify that the overbar denotes a time mean (rather than an area mean). With these edits, the interpretation of the equation is hopefully clear.

(b) If this equation were summed over all 42 cloud types, would that equal the overall gridbox CRE, in a manner similar to what one gets for all-sky TOA flux in a gridbox (i.e., sum of Rclr plus the sum of f(p, tau)*Rcld(p, tau) for all (p, tau) cloud types)? I wonder if substituting L with Ln means there is a lack of closure between the gridbox CRE and the sum of CREs from all individual cloud types. One can test this by checking if CRE – RSWlow equals RSWhigh calculated from individual cloud types in p=3 to 7, where CRE is the gridbox cloud radiative effect. If these are not consistent, what are the implications for how we interpret trends in RSWlow?

Physically, differences between L and $L_n$ reflect variations in upper-level cloud amount, which obscure low-level clouds by an amount that varies in time. Therefore, substituting $L_n$ for L in the equation quantifies the radiative impact of obscuration. This obscuration term needs to be re-assigned to upper-level cloud-radiative anomalies in order to obtain CRE closure, as described in Zelinka et al. 2025 (10.5194/acp-25-1477-2025).

For our analysis, the trend contribution from obscuration term is small: we have verified this by re-computing SWCRE$_{low}$ without accounting for obscuration by upper-level clouds (Fig. R2 below). Without the obscuration correction, the SWCRE$_{low}$ trend increases by just 0.007 W m$^{-2}$ decade$^{-1}$, which we feel is small enough to have no impact on the interpretation of the findings. This is now noted in the text (L235):
"Note also that accounting for obscuration by upper-level clouds, as done here, has little impact on the results, decreasing the SWCRE$_{low}$ trend by just 0.007 W m$^{-2}$ decade$^{-1}$ (not shown)."

[Figure]

*Fig. R2. Timeseries of SWCRE$_{low}$ with (black) or without (red) correcting the cloud amount data for obscuration by upper-level clouds.*

Lines 226-228: "Finally, we concatenate the Terra–Aqua data up to February 2023 with the modified NOAA-20 data from March 2023 to July 2024, to produce a single record from September 2002 to July 2024."
Why transition to NOAA-20 in February 2023? The Loeb et al. (2024) paper makes the transition to NOAA-20 in April 22 to avoid any impact of Terra and Aqua orbit drift. How does the date of the transition impact the trend?
We had missed that Loeb et al. transition in April 2022, thanks for pointing this out. Having checked, using this earlier transition does not affect the SWCRE$_{low}$ trend, which remains at 0.22 W m$^{-2}$ decade$^{-1}$. The revised plots use values based on the April 2022 transition, and the text here has been revised accordingly.

Line 230: "Ceppi et al. (2024), with the addition of an aerosol CCF following Wall et al. (2022)."
A more recent study by Park et al (2024; https://doi.org/10.5194/egusphere-2024-2547) argues that the activation rate of cloud droplet number concentration in response to variations in aerosol (e.g., sulfate aerosols) needs to be explicitly accounting for, otherwise the influence of aerosol is overestimated. At the very least, this possibility should be acknowledged.
We agree that this needs to be explicitly acknowledged, although we think the jury is still out as to whether accounting for aerosol activation leads to more accurate estimates. We have added the following sentence at the end of the paragraph (L266): "Note also that we do not account for potential effects of incomplete activation of aerosol droplets, which would likely yield a smaller estimate of the aerosol effect (Park et al., 2025)."

Line 231: "SW low-cloud radiative anomalies"
Shouldn't this be "SW low-cloud CRE anomalies" instead?
This has been modified to "the low-cloud SWCRE anomalies at each location r, dSWCRE$_{low}$(r)…"

Lines 258-259: "This assumes that the rate of global warming is forced on this 21-year timescale, and that the CCF forced responses scale with global warming."
Is there any justification for using this assumption?
Please see point 2 of the summary of main changes on page 1 of this document. We now account for uncertainty in the estimate of the model-derived forced and unforced components of the CCF

and SWCRE trends, as explained in the revised Appendix A5. Furthermore, we no longer make the assumption that the CCF forced responses scale with global warming – and therefore no longer rescale the model-derived forced CCF trends by the ratio of observed to model-simulated global warming. By dropping this assumption, we account for uncertainty in both the magnitude and pattern of the forced responses. The impact of including this additional uncertainty is overall small, widening the $\pm 1\sigma$ ranges by less than 0.01 W m$^{-2}$ decade$^{-1}$.

Figure A3: Please indicate what "log(s)" is in the caption or main text (e.g., lines 180-182). It is only revealed that s is sulfate mass concentration on line 324, well after the figure is introduced. Apologies for the oversight. Since log(s) is now introduced in section A1 along with all other CCFs, we believe it is no longer necessary to define this term here.